# Hierarchical ground-state crystals underlying Hertzian quasicrystals

Yao Li [1] ✉, Yiwei Wang[1], Yingke Geng[1], Wenyu Liu[1], Fangfu Ye[2,3] ✉ &
Jeff Z. Y. Chen [4] ✉

As a simple physics model for crystal and quasicrystal formation, the Hertzian potential describes the interaction between two soft-core colloidal particles. However, explaining why quasicrystals emerge in such a minimal model remains a theoretical challenge. Here, using analytically exact approaches, we reveal multiple layers of hierarchical crystal patterns that serve as the energy ground states underlying quasicrystals found by computer simulations. Our findings offer a new perspective on quasicrystal formation.

A typical, idealized crystal contains a repeatable basic unit cell that is periodically arranged in space. A quasicrystal retains an orientational symmetry, but lacks translational periodicity[1,2]. Discovering the physical mechanisms behind the quasicrystal structures is an actively ongoing research field in physics. Many can be identified with the mechanism that two or more competing length scales are required in the system, e.g., caused by mixing two constituent particle types. Well-known examples include many binary alloys[3], designed two-length-scale materials[4–8], and anisotropically shaped molecules[9–14]. Investigating the origin of quasicrystals is one of the central themes in materials science[15,16], since the very beginning of the discovery, still far from a complete understanding[17,18]. An interesting comparison can be drawn from architectural patterns in many existing cultures[19].

Recent decades have witnessed the interest in a new mechanism of self-assembly: packing identical soft-core particles can produce a surprisingly rich variety of crystal structures[20–26] and, under certain conditions, even quasicrystals[23,25]. A well-known example is the Hertzian interaction, derived from classical elasticity theory[27], which provides one of the simplest models of a soft-core potential. This unique class possesses a strikingly simple mechanism that can be realized in many practical systems[26,28–30]. Yet, the quasicrystals observed in Hertzian particle systems[25,31] remain puzzling.

Here, we explore the origin of hierarchy-built crystals and quasicrystals formed by Hertzian particles discovered in this work and reported in the literature[25,31], and their relationship to the five common crystal structures. The theoretical study carried out is based on a system of particles, in which any two at a distance $r$ interact with each other through the generalized Hertz potential

$$u(r) = \begin{cases} \varepsilon(1 - r/\sigma)^{\alpha} & \text{(when } r < \sigma), \\ 0 & \text{(otherwise)}. \end{cases} \quad (1)$$

The parameters $\varepsilon$ and $\sigma$ define the units of energy and length, respectively. The parameter $\alpha$ controls the softness of the interaction. The reduced temperature and reduced density are defined as $T^* = k_B T/\varepsilon$ and $\rho^* = \rho\sigma^2$, respectively, where $\rho$ is the number density of particles. The ground states of the system, when $T^* \to 0$, is completely controlled by two dimensionless parameters, $\alpha$ and $\rho^*$.

## Results

Previous computer simulation studies have been carried out to determine the ground-state phase diagram, which includes basic crystalline states such as triangle (Tri), square (Sq), stripe (Str), stretched honeycomb (SHon), and honeycomb (Hon), located in the uncolored region of Fig. 1. The proximity of the colored region was thought of as the location where a quasicrystal (QC) emerges[25].

From the current study, however, this is not the case, when crystal patterns composed of complicated tilings of weakly deformed pentagons and other geometric shapes are carefully studied in this parameter regime. Here, from an analytical approach, we calculated the energy minima corresponding to Pen-Sq (Cairo), PenTri-Tri (Chengtu), PenHex-Str, PenHex-Sq (Agra), and PenTri-Rec crystals illustrated in Fig. 1B–F. It is discovered that they all have a lower system energy than that of the previously proposed QC state. The name of each crystal is

[1]School of Physics and Key Laboratory of Functional Polymer Materials of Ministry of Education, Nankai University, and Collaborative Innovation Center of Chemical Science and Engineering, Tianjin, China. [2]Beijing National Laboratory for Condensed Matter Physics and Laboratory of Soft Matter Physics, Institute of Physics, Chinese Academy of Sciences, Beijing, China. [3]Oujiang Laboratory (Zhejiang Lab for Regenerative Medicine, Vision, and Brain Health), Wenzhou Institute, University of Chinese Academy of Sciences, Wenzhou, Zhejiang, China. [4]Department of Physics and Astronomy, University of Waterloo, Waterloo, ON, Canada. ✉e-mail: liyao@nankai.edu.cn; fye@iphy.ac.cn; jeffchen@uwaterloo.ca

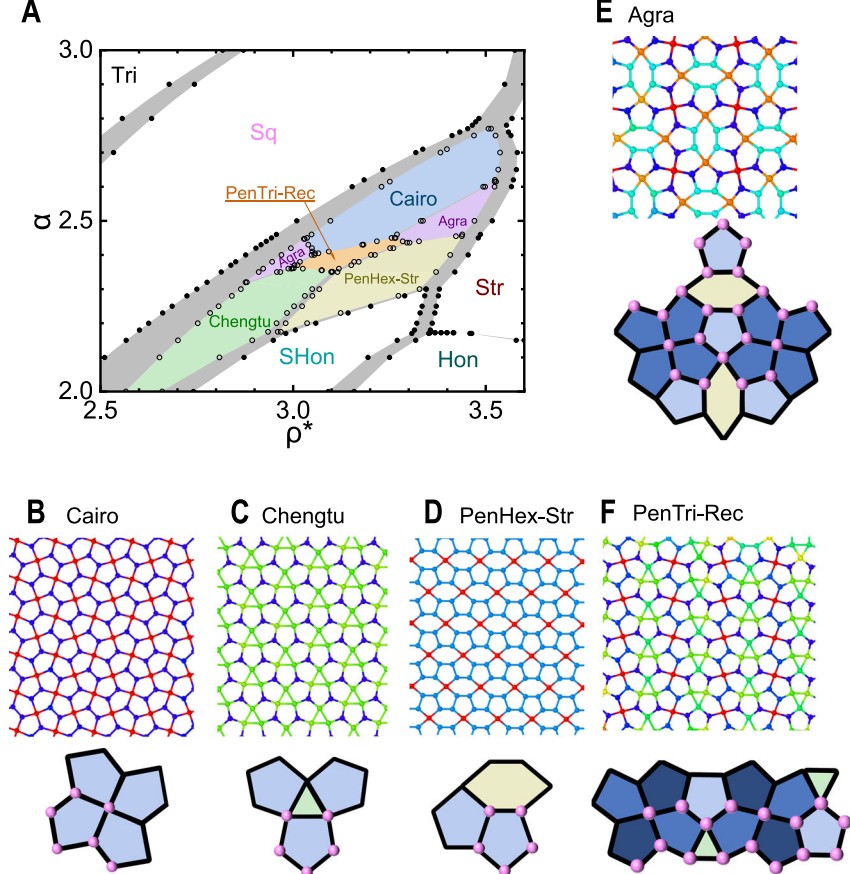

**Fig. 1 | Phase diagram and crystalline structures related to the quasicrystal state. A** Two system parameters $\alpha$ and $\rho^*$ span the phase diagram, determined from exact treatment of the system potential energy. Those with basic tilings are uncolored: Tri (triangle), Sq (square), Str (stripe), SHon (stretched honeycomb), and Hon (honeycomb), as well as those with sophisticated, pentagon-related tilings are colored: Cairo, Chengtu, PenHex-Str, Agra, and PenTri-Rec. The structures of the latter are further displayed in (**B**–**F**) in two views: in long-range order (upper panels) and their primitive cells (low panels) where solid lines and pink spheres represent tiles and particles in each primitive cell. In each tiling, congruent tiles are colored with the same color. The gray zones in (**A**) represent coexistence of nearby phases.

separated by two parts, of which the left part lists its composing polygon(s), and the right one denotes its super-lattice. The study is based on an analytical and numerical computation of multiple energy branches, each corresponding to a given periodic crystal structure. A standard tangent construction determines the region of a stable phase on the phase diagram spanned by the parameter pair $[\alpha, \rho^*]$, for the first-order phase transition between them. The narrow shaded corridors in Fig. 1A are the region where the adjacent phases coexist.

By searching the pattern library, we found that the patterns in Fig. 1B, C, E correspond to those documented in historical cultures, whereas PenHex-Str and PenTri-Sq show no obvious correspondence. The well-known Cairo lattice (Fig. 1B) appeared on the streets of Cairo and in many other Islamic decorations[32]. The second phase (Fig. 1C) takes the name of Chengtu tiling, as similar patterns appear on the armor of door gods and in traditional window grilles in Chengtu, China[33,34], both following the same $p31m$ symmetry group. A tiled superlattice of the $p31m$ symmetry was also reported in ref. 35. The Agra tiling, for the pattern in Fig. 1E, exhibits $p4gm$ symmetry and resembles a stone decoration found on monuments in Agra, India[36,37].

The Cairo pattern was realized in materials science by modifying the normal graphene[38] with hexagon cells and by connecting carbon pentagons together in graphenes[39]. The successful synthesis of hundreds of other crystals with similar patterns was made, showing innovative physical properties[40]. A crucial point is that it was previously unknown whether the Cairo lattice could be formed by

isotropic particles interacting via, for example, the Hertzian potential, shown here.

Separately, in mathematics, the problem of tiling a plane with monohedral convex pentagons has attracted extensive study for over a century. Only very recently has it been shown that exactly 15 types of such tilings exist[41,42], all of which were exhaustively considered in our study. For instance, Cairo tilings fall into both type 2 and type 4 classifications.

The PenHex-Str pattern is similar to those previously reported for 2D particles in a theoretical analysis of a hard-core potential decorated by a linear ramp[43], and in a computer simulation of a hard-core square-well potential[44]. These studies found related tenfold and fivefold quasicrystals respectively, other than the twelvefold quasicrystals in the current study.

The main symmetry properties of all five crystal structures found in this study are summarized in Table 1 and supplementary Information (Supporting materials online). We now highlight the relations among these pentagon-involved phases, which can be classified into three hierarchies. The Cairo pattern, Pen-Sq, is a monohedral pentagon only tessellation, which is classified in the basic hierarchy I. Hierarchy II contains both Chengtu and PenHex-Str, which are binary tessellations of pentagons and triangles, pentagons and stretched hexagons respectively. Note the triangles and the stretched hexagons are the basic tiles in the simple lattices, Tri and SHon, respectively (see "Methods").

**Table 1 | Main symmetry properties of the five pentagon-based tilings and the dodecagonal quasicrystal (DDQC)**

| Tiling name | Cairo (Pen-Sq) | Chengtu (Pen-Tri-Tri) | PenHex-Str | Agra (PenHex-Sq) | PenTri-Rec | DDQC (PenTriHex-Ap) |
|---|---|---|---|---|---|---|
| Pentagon | 4 | 3 | 2 | 12 (4 + 8) | 10 (2 + 8) | Yes |
| Regular triangle | — | 1 | — | — | 2 | Yes |
| Hexagon | — | — | 1 | 2 | — | Yes |
| Vertices | 6 | 5 | 5 | 22 | 16 | ∞ |
| Hierarchy | I | II | II | III | III | — |
| Super-lattice (Bravais) | Square | Triangle | Centered rectangular (Stripe) | Square | Rectangular | — |
| Wallpaper group (IUCr†) | *p4gm* | *p31m* | *c2mm* | *p4gm* | *p2mg* | — |
| Rotational symmetry (-*fold*) | 4 | 3 | 2 | 4 | 2 | 12 |

The first three rows list the number of basic lattice units and the fourth the vertices *N*, per super-lattice unit cell. The fifth row and beyond list the basic super-lattice symmetry, which further breaks down to the number of bilateral symmetry axes and the global rotational folds of the super structure. Note that, the least common multiple (lcm) of the first five numbers in the last row, lcm(4, 3, 2, 4, 2) = 12, is exactly the symmetry fold of a DDQC. The shapes of involved pentagons are all bilaterally symmetric, except for those in both Agra and PenTriRec, where the eight pentagons contain two weakly deformed Cairo 4-prototiles.
†The IUCr notation: the notation for the symmetry group adopted by the International Union of Crystallography.

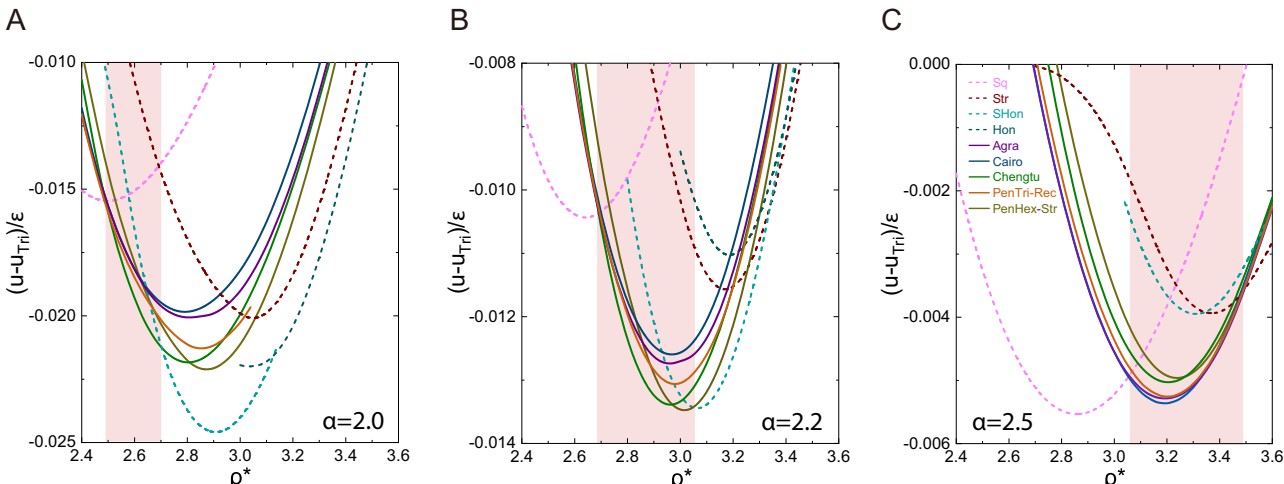

**Fig. 2 | Multiple energy branches, $U - U_{Tri}$, for the three typical cases of $\alpha$.**
**A–C** These mean energies per particle are calculated after minimization to attain the optimal configurations, based on an analytical approach. The colors used for the energy plots match those used in Fig. 1A to label the states, and are also indicated in the legend of (**C**). For easy recognition, the solid curves are for the pentagon-involved phases, and the dashed curves are for the other simple phases, Sq, Str, SHon, and Hon. The shaded ranges highlight the approximate density regions where pentagon-involved phases are identified through a double-tangent construction. The energy of the triangular lattice $U_{Tri}$ is taken as the reference.

The other two phases, Agra and PenTri-Rec, are in hierarchy III. The primitive cell of the Agra lattice can be viewed as the superposition of two Cairo primitive cells and two PenHex-Str primitive cells in a cross arrangement. By the same token, the primitive cell of PenTri-Rec can be recognized as the superposition of two pairs of Cairo and Chengtu primitive cells. These four proto-cells are linear-alternately arranged as Cairo|Chengtu|Cairo|Chengtu, where one Cairo|Chengtu is the vertical reflection of the other. A lattice at a higher hierarchical level is built upon ones at lower levels. Even more complex hierarchical phases could be built, which warrants future exploration.

The curves in Fig. 2 represent the mean energy per particle, after the lattice configurations are analytically optimized with respect to the side lengths of a given geometry, to yield the system energy minimum (Supporting materials online). Within the highlighted regions where the five pentagon-related states are stable, the energy branches of these states are very close to each other, in comparison with the distinctively much higher energies of other states. The analytical tool enables us to find the fine energy differences between these states,

which is critical for performing the tangent construction to determined the first-order transitions in Fig. 1.

On the other hand, the aperiodic nature of a quasicrystal prevents us from conducting an analytical analysis. To study the quasicrystalline formation, molecular dynamics (MD) simulations were used, which are coupled with a simulated annealing schedule to lower the system temperature. During the annealing processes, a dodecagonal quasicrystal (DDQC) appears first after solidification. Examples are displayed in Fig. 3A where the black circles represent the DDQC energy scale and a typical snapshot is shown in panel B. Upon further freezing, the system transitions to either a phase-separated structure that contains co-existing domains or a single crystalline, exhibiting the pentagon-related states in Fig. 1B–F. The DDQC content approaches 0, shown in Fig. 3A by the diamonds, now with a lower energy. A snapshot of twinned crystallines is shown in panel D.

Over most of the colored phase regions in Fig. 1, MD runs are typically trapped energetically to contain pentagon-related crystalline domains in different orientations[3,15], because of the closeness of lattice

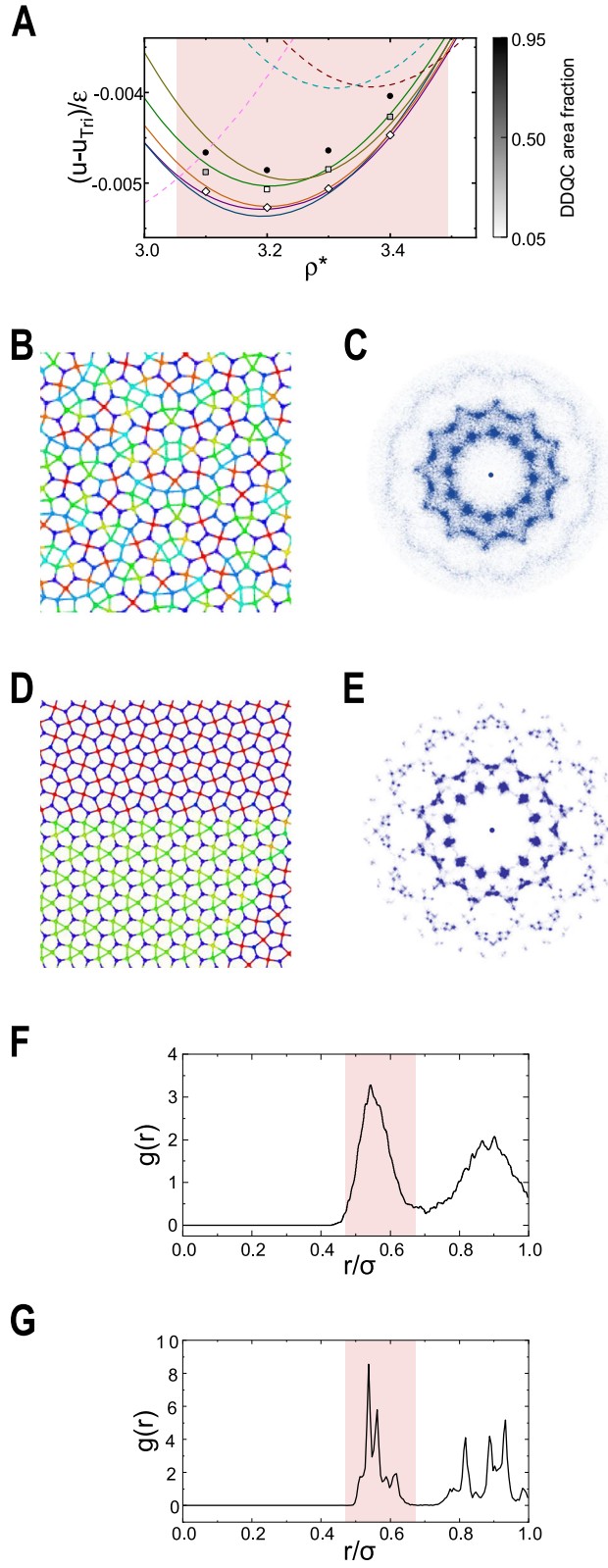

**Fig. 3 | DDQC and co-existing crystallites.** The mean MD simulation energies for $\alpha = 2.5$ are shown in (**A**), with $T^* = 4 \times 10^{-4}$ (circles), $2 \times 10^{-4}$ (squares), and $1 \times 10^{-4}$ (diamonds). The grayscale indicates the DDQC area fraction, and the meanings of the solid and dashed lines are the same as in Fig. 2. A DDQC is displayed in (**B**), with its calculated diffraction pattern in (**C**). The twinned Cairo and PenHexStr domains are clearly visible in (**D**); its diffraction pattern in (**E**) shows sharp spots characteristic of the twinned crystalline structure. The radial distribution functions of the DDQC in (**B**) and the crystalline domain in (**D**) are shown in (**F**) and (**G**). The first hump in (**F**) splits into four peaks in (**G**) in the shaded range. The fine peak positions correspond to the analytically determined edge lengths of Cairo and PenHex-Str.

and boundaries. Hence, the DDQC is preferred and stabilized by the entropy. As the entropic contributions to the total free energy vanish when the system temperature approaches absolute zero, the system settles for the more ordered crystalline ground states shown in Fig. 1. This scenario agrees with the random-tiling hypothesis[16,45]. The energy scales of intermediate stages are shown in Fig. 3A by gray squares.

Is the 12-fold rotational symmetry in the diffraction pattern, clearly seen in Fig. 3C as the characteristic of DDQC, a unique signature that can be used for its identification? Here, we show that a co-existing crystalline structure produced from MD can also produce the same symmetry. Note that the last row of Table 1 summarizes the rotational symmetries of the related states; DDQC's 12-fold symmetry is the least common multiple of all other listed crystals, revealing the underlying relationship to these crystals. Take the example of the diffraction pattern of the Cairo and PenHex-Str co-existence (Fig. 3D) shown in Fig. 3E. It also possesses the 12-fold symmetry, but each peak contains a fine structure implying the formation of crystal order. The radial distributions $g(r)$ of DDQC and the co-existence structure are shown in Fig. 3F and G. The first hump in $g(r)$ of DDQC is wide and diffuse, while in the co-existence structure, it is composed of four sharp fine peaks, of which two are the edge lengths of the Cairo lattice and the other two are those of the Chengtu. Hence, the 12-fold diffraction symmetry needs to be examined carefully.

A fundamental theoretical approach to understanding quasicrystal formation has involved the study of complex crystals—termed crystalline approximants—that retain large periodicity while exhibiting analogous structural motifs[15]. In this work, we report the exactly-solved family of pentagon-tiled crystals as zero-temperature ground states. The quasicrystal configurations emerge only at finite temperatures and do not represent the ground state. Earlier studies, using models such as the hard-core linear ramp[43] or the hard-core square-well potential[44], recovered at most a single complex crystal as the ground state. Sequences of approximants were observed only as entropy-stabilized phases, appearing upon heating, e.g., in a study of a 2D system with Lennard-Jones-Gauss interactions[35,46]. Here, the finding of ground-state hierarchies therefore provides concrete evidence that redefines our understanding of quasicrystals and their so-called approximants. It also raises further questions: How do these ground-state hierarchies evolve upon heating until quasicrystalline order emerges? Do additional higher-order crystals appear at intermediate temperatures of the currently studied system? Both issues deserve further exploration.

In summary, we have provided analytical and simulation data to address a fundamental question: what are the low-temperature crystal and quasicrystal states, and how are they related to each other? A perceived QC can eventually yield to the formation of hierarchical crystalline states at low temperatures. The existence of other pentagon-related patterns demonstrated here beyond Cairo suggests the possibility of new synthetic materials. The Hertzian model used here serves as a tool to illustrate the underlying physics. Whether multiple hierarchies exist universally, and what kinds of hierarchical unit cells are fundamental in other potential-energy models (and even in nature), remains an open and interesting question.

energies. The pentagon-related crystals and DDQC are all composed of different mixtures of pentagons and other geometric entities, in various tiling patterns. At a low but nonzero temperature, the entropy is low enough to accommodate DDQC, yet high enough to surpass the minuscule energy differences between these pentagon-related crystals

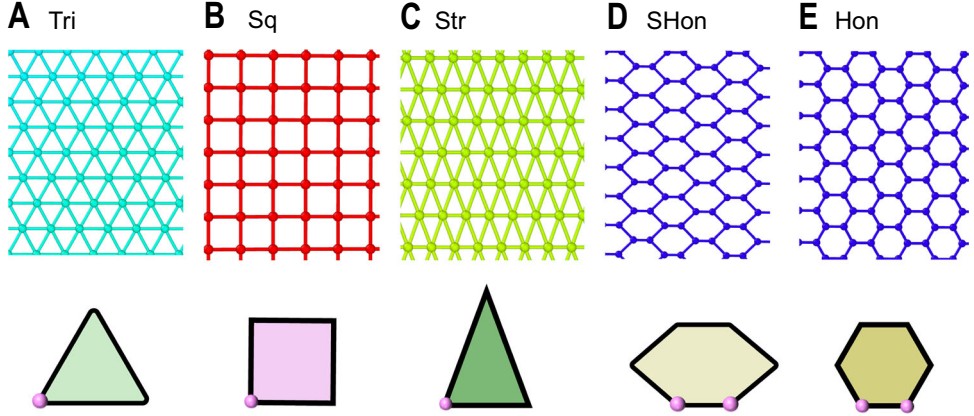

**Fig. 4 | Other ground-state crystalline structures. A–E** Long-range structures (upper panel) and the unit cells (lower panel) used in calculating these ground states: Tri, Sq, Str, SHon, and Hon.

### Table 2 | Main symmetry properties of other simple crystals

|      | Vertices | Wallpaper Group (IUCr) | Super-lattice (Bravais) | Rotational symmetry (-fold) |
|------|----------|------------------------|-------------------------|------------------------------|
| Tri  | 1        | *p6mm*                 | Triangular              | **6**                        |
| Sq   | 1        | *p4mm*                 | Square                  | **4**                        |
| Str  | 1        | *c2mm*                 | Centered rectangular    | **2**                        |
| Hon  | 2        | *p6mm*                 | Triangular              | **6**                        |
| SHon | 2        | *c2mm*                 | Centered rectangular    | **2**                        |

## Methods

### Crystal states and minimization procedure

For the crystal states, the reduced mean energy per particle is defined by

$$\frac{U}{\varepsilon} = \frac{1}{2N} \sum_{i=1}^{N} \sum_{j} u(r_{ij}) \tag{2}$$

where $N$ is the number of particles in the primitive unit cell, and $j$ runs through all particles within the force range $\sigma$ in the Hertzian potential. And the summation runs over all vertices of the super unit cell. The unit cells used for PenHex-Str, PenTri-Rec, Cairo, Chengtu and Agra are illustrated in Fig. 1 of the text, and for Tri, Sq, Str, SHon, and Hon in Fig. 4 and Table 2.

Take the geometry of the Chengtu pattern for example. The congruent pentagons of the hierarchical tilings are assumed bilaterally symmetric, with four sides having equal length $a$ and one side having a different length $b$, sharing with the regular triangular side length. Required by symmetry, the two connected pentagon shoulder angles are fixed at $\pi/3$. The minimization is then taken with respect to $a$ and $b$, in consideration of a fixed $\rho^*$ as a constraint. More detailed derivations can be found in Supplementary Information (Supporting materials online).

### Molecular dynamics and simulated annealing

This technique is mainly used for minimization of the energy, expressed in the last section to yield DDQC and crystalline structures at ultra-low temperatures. One takes a system of $N$ particles and minimizes the system energy by bringing the simulated temperature down to a low, nonzero temperature in a computer simulation. A system of $N = 4 \times 10^3$ particles is prepared in an area $A$, which defines the density $\rho^*$. A standard LAMMPS package[47] is used in a prescribed simulated annealing protocol. The annealing starts with the temperature $k_B T/\epsilon = 0.1$. Every $2 \times 10^4$ time steps the temperature is lowered by multiplying the factor $1 - 3 \times 10^{-4}$, until $k_B T/\epsilon$ is below a targeted low temperature ($4 \times 10^{-4}$, $2 \times 10^{-4}$, or $1 \times 10^{-4}$ in Fig. 3).

To minimize the effects of boundary conditions, we varied the system size using three values $N = 2 \times 10^3$, $4 \times 10^3$, and $10^4$; the radius $R$ was adjusted accordingly to maintain the same $\rho^*$ value. The variations of the produced energies, in particular for the data points plotted in Fig. 3A, are smaller than the size of the plotted symbols. Therefore, finite-size effects have a negligible impact on the DDQC content analysis presented in that figure.

All regular crystalline structures are calculated analytically, as explained in the last section. LAMMPS[47] has been used to verify that the analytic result is correct, after lowering the temperature below $1 \times 10^{-5}$.

### Diffraction pattern

The diffraction patterns are the intensity maps of the static structure factor: $S(\boldsymbol{k}) = \frac{1}{N} \langle \rho(\boldsymbol{k})\rho(-\boldsymbol{k}) \rangle$ where $\rho(\boldsymbol{k}) = \sum_{i=1}^{N} e^{i\boldsymbol{k}\cdot\boldsymbol{r}_i}$ is the Fourier transform of the number density of the particles, where $\boldsymbol{r}_i$ is the coordinate of particle $i$. The diffraction pattern images are generated using the software SingleCrystal®.

## Data availability

The source data for Figs 1–3 and other raw data are deposited in Zenodo under accession code https://doi.org/10.5281/zenodo.17474851.

## Code availability

The simulation codes used in this study are deposited in Zenodo under accession code https://doi.org/10.5281/zenodo.17474851.

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

## Acknowledgements

This work was supported by the National Natural Science Foundation of China (12275137 granted to Y.L.) and Nature Science and Engineering Council of Canada (granted to J.Z.Y.C.). We thank the Digital Research Alliance of Canada for providing computational resources.

## Author contributions

Y.L. designed the project. Y.L., Y.W., and Y.G. performed the theoretical calculations and simulations. Y.L., Y.W., Y.G., W.L., F.Y., and J.Z.Y.C. analyzed the data and wrote the manuscript.

## Competing interests

The authors declare no competing interests.
