## [Transparent Peer Review file · Nature Communications]

Hierarchical ground-state crystals underlying Hertzian Quasicrystals

Corresponding Author: Professor Yao Li

Version 0:

Reviewer comments:

Reviewer #1

(Remarks to the Author)

Ref.25 (Nat. Commun. 8, 2089) reported the density-alpha phase diagrams with a dodecagonal quasicrystal (DDQC) phase at a low temperature where alpha is a parameter in the Hertz potential. This manuscript gives a more precise density-alpha phase diagram at zero temperature. The phase diagram shows that the quasicrystal regime in ref.25 contains 5 types of quasicrystals instead of one type DDQC in ref.25; moreover, the authors analytically calculate the total energy of each type of perfect quasicrystals and claim that they are lower than that of the DDQC. They show that DDQC is likely to be a mixture of some of the 5 types of quasicrystals and can exist at finite temperatures. This manuscript clarifies the ground-state quasicrystal phases in the 2D Hertzian system. I feel that it can be published in Nat. Commun. after the authors properly address the following questions.

Which figure/paragraph support the statement “we discovered crystal patterns built hierarchically behind quasicrystals” in the abstract?

Page 1 “all have a lower system energy than that of the previously proposed QC state”. However, I cannot find the value of the energy of the DDQC. Although it cannot be analytically calculated due to its aperiodic nature, the author should provide the measured energy from the simulation to demonstrate that it is indeed higher than the energies of the other five types of quasicrystals.

By heating a ground state to the low temperature used in ref.25, does the ground state transform to DDQC?

Page 1 “since the very beginning of the discovery (17, 18)”. Refs.1,2 are the very beginning of the discovery, why cite 17,18?

A reference should be added after “... in terms of [alpha rho]” in page 1.

Reviewer #2

(Remarks to the Author)

The manuscript NCOMMS-24-02433-T “Premodern Architectural Tilings behind Hertzian Quasicrystals” by Li, Ye, and Chen, which reports a numerical study of 2D crystalline and quasicrystalline structures formed by particles characterized by the Hertzian pair interaction, cannot be recommended for publication for several reasons as elaborated below, and I suggest it be rejected.

Firstly, the manuscript is hardly novel enough to be suitable for a high-impact journal such as Nat. Commun. The model explored has been studied in the past (e.g., in Ref. [31] and in related papers by Ryzhov, Fomin, Tsiok, and Gaiduk as well as by other authors). Li, Ye, and Chen do seem to extend the existing insight but even if they do so, the their results are incremental.

Secondly, my concern behind the wording “seem to extend” in the previous sentence pertains to the possibility that the analysis in NCOMMS-24-02433-T may not be scientifically sound. It appears to me that the authors constructed the phase diagram in the (α , ρ) plane in Fig. 1A by identifying the minimal-energy structures rather than by considering the

complete phase equilibrium, that is by the Maxwell double-tangent construction. I would imagine that the phase diagram in the (α , ρ) plane should feature finite coexistence regions (especially in view of the very small energy differences reported in Fig. 2) but they seem to be absent. If the authors indeed did not do the double-tangent construction, it is possible that some of the claimed stable structures would not make it in the correct phase diagram.

Thirdly, I do not share the authors' view that the Hertzian interaction is a good model for the pair potential between soft particles. The pairwise additive Hertzian interaction applies to contact repulsion between elastic bodies such as spheres but only for small indentations, that is for r/σ not much smaller than unity. The authors use it for all r 's, that is well beyond the applicability of this model to real elastic bodies. In addition, it is questionable that soft colloidal particles can be described by the Hertzian potential in the first place as they are too small for the theory of elasticity to be relevant. These concerns make NCOMMS-24-02433-T less relevant for real colloidal systems.

Finally, the presentation of the results is not commensurate with the results themselves; in brief, the results are oversold. The references to architecture, art, history, etc. are superficial and hardly needed.

Reviewer #3

(Remarks to the Author)

The paper discusses the formation of complex crystals by employing Hertzian potential energy, building upon the framework presented in Ref. 25 (Nat. Commun. 2017). The authors delve into the intricate crystal patterns associated with dodecagonal quasicrystals. The primary contribution of this paper lies in introducing premodern tilings. However, I would hesitate to argue that the paper warrants publication in Nature Communications.

Comments:

1. The primary outcome revolves around comparing the ground state energies of the new tilings. Regarding free energies, the Frenkel-Ladd method proves to be valuable for analyzing periodic crystals and has previously been applied to colloidal systems.
2. By using numerical simulations, the authors have not obtained any single phases of new tilings. They may depend on boundary conditions. I recommend that much numerical evidence supporting the main text be included in the Supporting Materials.
3. In the case of these complex crystal lattices, I presume that the minimization processes are multidimensional. However, the explicit process, including the number of parameters considered, is not delineated in the supporting materials. It may be assumed that symmetry considerations play a role in the process.
4. The Cairo tiling is well-known; however, I disagree with the authors' choice of terminology for the Chengtu and Agra lattices, despite their careful justification. To my knowledge, the names Chengtu and Agra lattices are not commonly used to refer to the lattice structures presented in the paper. It should be noted that the Chengtu tiling depicted in Fig. 1C appears to consist of equilateral tiles, including an equilateral triangle and an irregular but equilateral pentagon. However, the pattern described in Ref. 39 (Dye, Fig. C 5b) does not appear to be equilateral. Consequently, I am hesitant to adopt this terminology, even though the wallpaper group symmetry is consistent. Additionally, for the Agra tiling shown in Fig. 1E, I was unable to locate a reference. While I acknowledge the abundance of beautiful patterns in Agra, I am uncertain about using the Agra lattice to describe this specific and highly complex pattern. Instead, I suggest that the authors consider employing terminology from the theory of mathematical tiling.
5. The authors made assumptions regarding models, including the Cairo, Chengtu, PenHex-Str, Agra, and PenTri-Rec tilings. However, these assumptions lack systematic consideration. I would suggest that the authors consult the theory of tilings for a more rigorous foundation. Alternatively, the hierarchical or stepwise construction employed in this study recalls the approximant phases of quasicrystals.
6. For the names of the wallpaper (plane) group, the authors use the shorter versions. It would be better to use the formal (IUCr) version; say "p4g" is "p4gm." You may find the correspondence in a table in Wikipedia "wallpaper group".
7. The Chengtu tiling has been deeply studied in a PRL (2011) paper in the context of colloidal particle simulations. DOI: 10.1103/PhysRevLett.106.095504
8. Figure caption Fig.3 C Cairo and PenHex-Str? -> Cairo-Chengtu
9. PenHex-Str (Fig) vs PenHex-Rec in the main text (L124, L133)
10. L.163 3D -> 3C

Finally, I truly enjoy reading the manuscript.

Reviewer #4

(Remarks to the Author)

In this manuscript the authors determine the ground state (temperature $T \rightarrow 0$ limit) structures for a system of particles interacting via a generalised Hertz potential. At finite temperatures, this system was previously reported in Ref.[25] to form quasicrystals, but here they report that a number of periodic structures instead are the ground state at $T=0$. This is an important finding, if true. Particularly because some of the observed structures are visually appealing and have been used in historical architectural decorations.

However, I'm concerned that the way the energies for each structure are calculated is artificially penalising the candidate quasicrystalline structures that are actually observed at finite temperature. The energy of a quasicrystalline structure depends significantly on the box size in which one calculates. Remember, strictly speaking, one can never find a quasicrystal in a finite size box. There are methods to get around this for practical calculations. One approach is to work with a higher dimensional periodic systems and then cut and project to obtain the real lower dimensional system. Typically, one looks at a 4-dimensional periodic structure to calculate for the real 2-dimensional system. Another approach is to consider what finite sized box leads to a structure which forms a good finite size approximant to the true infinite quasicrystal. I saw no discussion about varying the size of the box used for the simulations of the quasicrystal nor any discussion of how the box size was chosen. I'd also like to see how the quasicrystal free energy varies for different choices of box when plotted on top of the curves displayed in Fig.2. Is the quasicrystal energy really nowhere near the energy of the periodic structures displayed?

While I agree the connection to patterns in architecture is interesting, this has been discussed before. I feel that the connections between the model pair potential and the particular structures formed and why they arise in historical building decorations needs to be drawn out better.

The correlation functions $g(r)$ in Fig.3 are interesting but the text does not really explain why these particular examples have been selected, nor what we are supposed to learn from these. These need to be discussed more. Ideally, I'd want to see how they vary as parameters in the model are varied.

A final, minor point: I don't really understand the use of the word "behind" in the title.

Version 1:

Reviewer comments:

Reviewer #1

(Remarks to the Author)

The authors have addressed my questions and comments satisfactorily. However, the caption of the newly added Fig. 3A should be improved, e.g., explaining the meaning of the solid and dashed curves.

Reviewer #2

(Remarks to the Author)

Reviewer #2:

I have read the revised version of NCOMMS-24-02433-A "Premodern Architectural Tilings and Hertzian Quasicrystals" by Li, Wang, Geng, Liu, Ye, and Chen as well as the rebuttal letter. The authors made some changes to the manuscript but my reservations remain. I do not find the manuscript suitable for publication in Nat. Commun. and I suggest that it be rejected. Below please find a more detailed response.

AUTHORS:

The manuscript NCOMMS-24-02433-T "Premodern Architectural Tilings behind Hertzian Quasicrystals" by Li, Ye, and Chen, which reports a numerical study of 2D crystalline and quasicrystalline structures formed by particles characterized by the Hertzian pair interaction, cannot be recommended for publication for several reasons as elaborated below, and I suggest it be rejected.

1. "Firstly, the manuscript is hardly novel enough to be suitable for a high-impact journal such as Nat. Commun. The model explored has been studied in the past (e.g., in Ref. [31] and in related papers by Ryzhov, Fomin, Tsiok, and Gaiduk as well as by other authors). Li, Ye, and Chen do seem to extend the existing insight but even if they do so, their results are incremental"

REPLY: There may be some misunderstanding here. Yes, the model has been studied in the past, but using it to explore the formation of pentagon-related crystalline structures in connection with quasicrystals, through an exact treatment that allows for precise determination of the phase diagram, is completely new. The references mentioned, for example, concern different aspects of crystal formation that are unrelated to the pentagon structures studied here.

Reviewer #2

The question is whether the paper contains enough completely new results to warrant publication in a high-impact journal. In my opinion, the crystal structures that contain pentagonal motifs and the dodecagonal quasicrystal reported are a modest step ahead. Indeed, it is possible that crystal lattice containing pentagons were not reported in a 2D system of Hertzian particles so far but not all of them are completely new. For example, the PenHex-Str structure was discussed 27 years ago in 2D particles with hard core + linear ramp potential [E. A. Jagla, Phys. Rev. E. 58, 1478 (1998); structure S4 shown in Fig. 2 of Jagla's paper]; in this paper, the PenHex-Str lattice is located right next to the stretched honeycomb (Shon) and the stripe (Str) lattice, and not far from the Square (Sq) lattice – just like in NCOMMS-24-02433-A.

While the Hertzian potential is different from the potential used by Jagla in the above reference, they share the soft repulsive nature and so is it not surprising that they lead to a similar phase diagram. The “exact treatment that allows for precise determination of the phase diagram” mentioned by the authors in the rebuttal is not completely new either. In 1998, Jagla used the same approach but he noted that “other (more stable) structures may have been missed”. The structures reported in NCOMMS-24-02433-T may be some of those missed by Jagla. At the same time, it is quite possible that Li et al. may have missed one or more complex structures themselves. – Yet another old reference with crystals that include pentagonal motifs is A. Skibinsky et al., Phys. Rev. E 60, 2664 (1999); the potential used here is isotropic hard core + square well. I imagine that there exist other related references both in the field of colloids and in other fields, possibly older than Jagla 1998 and Skibinsky et al. 1999.

As per the quasicrystals reported in NCOMMS-24-02433-T at finite temperatures, I note that Jagla’s paper also contains a decagonal quasicrystal containing many regular pentagonal and stretched honeycomb local environments. The same is true for the Skibinsky et al. paper.

Hence my view that the results reported in NCOMMS-24-02433-T do not constitute a novel enough and important enough body of work to warrant publication in Nat. Commun.

AUTHORS

2. “Secondly, my concern behind the wording “seem to extend” in the previous sentence pertains to the possibility that the analysis in NCOMMS-24-02433-T may not be scientifically sound. It appears to me that the authors constructed the phase diagram in the (α, ρ) plane in Fig. 1A by identifying the minimal-energy structures rather than by considering the complete phase equilibrium, that is by the Maxwell double-tangent construction. I would imagine that the phase diagram in the (α, ρ) plane should feature finite coexistence regions (especially in view of the very small energy differences reported in Fig. 2) but they seem to be absent. If the authors indeed did not do the double-tangent construction, it is possible that some of the claimed stable structures would not make it in the correct phase diagram.”

REPLY: This is a very good point. Our original intention was to consider only the ground states (hence the old Fig. 1A was labeled as a ground-state diagram), with the discovered state contrasted against related papers where ground states were also considered. In this revision, we have transitioned to a true phase diagram, where the first-order transition lines are determined using the double-tangent construction. The revision preserves the overall structure of the original state diagram but enriches it by including coexistence corridors along the phase boundaries. Please see the revised Fig. 1A. We have also updated many parts of the text where “ground state” was originally referenced. We are deeply indebted to the referee for this insightful and helpful suggestion. The main physics remains the same.

Reviewer #2:

What thermodynamic potential was used for the double-tangent construction? Enthalpy or energy? Or was it the energy less the energy of the triangular lattice shown in Fig. 2? In my experience, the small energy differences shown in Fig. 2 (e.g., between Cairo in Agra at $\alpha = 2.5$) usually lead to broad regions of coexistence. From the data included in the manuscript, it is unclear how the authors constructed the phase diagram.

Fig. 2 still shows the range of densities where phases that include pentagons have the lowest energy (“The shaded ranges highlight the approximate density where pentagon-involved phases are preferred”), which is misleading.

AUTHORS

3. Thirdly, I do not share the authors’ view that the Hertzian interaction is a good model for the pair potential between soft particles. The pairwise additive Hertzian interaction applies to contact repulsion between elastic bodies such as spheres but only for small indentations, that is for r/σ not much smaller than unity. The authors use it for all r ’s, that is well beyond the applicability of this model to real elastic bodies. In addition, it is questionable that soft colloidal particles can be described by the Hertzian potential in the first place as they are too small for the theory of elasticity to be relevant. These concerns make NCOMMS-24-02433-T less relevant for real colloidal systems.

REPLY: The debate over whether the Hertzian model is valid for elastic particles lies beyond the scope of the current manuscript. In a separate study, we examined another well-documented pair potential and found that similar pentagon-related structures, as reported here, can also emerge. Albeit in different regions of the parameter space specific to that model, the pentagon-related crystallines are more common than one would think.

It is worth noting that the term “soft colloidal particles” is used here in a general sense; for example, they may refer to polymeric soft spheres of significantly larger size (see Ref. 21, 22 and 26 in the text), which have been modeled using the Hertzian potential. Notably, in Ref 26 (Grillo et al, Nature 2020) Hertzian model directly reproduced the experimental results. Indeed, recent studies have reported quasicrystal formation in polymer complexes. The question of whether “soft colloidal particles can be described by the Hertzian potential in the first place, given that they are too small for continuum elasticity to apply,” is itself open to debate. In this study, we use the Hertzian model, easy to adopt and conceptually simple, as a tool to explore the formation of pentagon-related structures. Our intention is not to apply the model strictly to the specific class of colloids the referee may have had in mind.

Reviewer #2:

In the first sentence of the abstract, the authors say “In its unique place as a fundamental physics model and corner stone supporting the current understanding of formation of crystals and quasicrystals, the Hertzian potential energy describes the interaction between two soft-core colloid particles.” In their response, the authors say that the relevance of the Hertzian

model is “open of debate”, which is evidently inconsistent with the first sentence of the abstract. In my view, the opening sentence of the abstract is exaggerated, because the Hertzian potential is not a cornerstone of the current understanding of formation of crystals and quasicrystals. Equally strong is the authors’ claim that the Hertzian interaction is “the prototypical interaction of a soft-core potential” (l. 31-33). I beg to differ. Why should the Hertzian interaction be the prototype? It seems to me that these statements are included so as to elevate the importance of the authors’ work; in my opinion, these statements are unjustified.

The Hertzian interaction is derived by assuming that the indentation of the particles in contact is small and that the deformation of the particles is described by the harmonic theory of elasticity. None of this holds at large indentations, and there exists no theoretical justification for the use of the Hertzian theory at indentations beyond a few percent. I do not see how this can be open to debate. In addition, the size of colloidal particles is typically no larger than a micrometer or else they sediment. It is hardly evident that particles as small as this can be regarded as elastic continua like macroscopic entities such as cm-size rubber balls.

AUTHORS

4. “Finally, the presentation of the results is not commensurate with the results themselves; in brief, the results are oversold. The references to architecture, art, history, etc. are superficial and hardly needed.”

REPLY. These references, depending on the audience and community, may hold value. For example, the Cairo tiling has traditionally been associated with a specific type of pentagonal tiling. While the scientific content of the manuscript would remain intact without these references, we believe that they offer useful context. We are open to removing them if necessary but would prefer to seek guidance from the journal’s Editor on this matter.

Reviewer #2:

A considerable part of the manuscript, especially in l. 95-123, refers to patterns used in architecture and for decoration. These references are episodic and serve no scientific purpose (e.g. “Dating back as early as 1800 A.D., the Chengtu pattern was scholarly documented by physicist D. S. Dye in his 1937 book (39), which recorded over a thousand types of Chinese lattice designs”); no conclusion is drawn based on the comparisons made. In their rebuttal, the authors say this themselves (“... the scientific content of the manuscript would remain intact without these references ...”). I do not share the authors’ view that “[these references] offer useful context”. They may appeal to a general audience and would possibly fit into a popular science article. In a research paper, on the other hand, they are unneeded or may even be a distraction.

In a similar vein, I find the title “Premodern architectural tilings and Hertzian quasicrystals” misleading. In the manuscript, the authors do not establish a connection between the architectural tilings and 2D structures formed by Hertzian particles (in my opinion, this cannot be done because such a connection does not exist). What is then the purpose of the title, which makes one expect the connection? The last sentence of the abstract reads “The connection between art history and the new crystal patterns can mutually stimulate the creativity of art designs and new material designs.” What exactly did the authors mean by this? Firstly, this connection does not exist, and secondly, if they meant the similarity of decorative and crystal patterns, this is no news; after all, 2D space groups are referred to as the *wallpaper* groups. There is no need to emphasize this in the abstract of a paper in Nat. Commun. in 2025.

The language is often poor – here are some examples:

l. 193-185: “The pentagon-related crystals and DDQC are all composed of different mixing of pentagons and other geometric identities, in various tiling patterns.” -> “The pentagon-related crystals and DDQC are all composed of different mixture of pentagons and other geometric entities, in various tiling patterns.”

l. 204-205: “underlining relationship” -> “underlying relationship”

l. 209: “the radius distributions” -> “the radial distribution function”

l. 224-225: “can stimulate the creativity of new material design” -> “can stimulate the design of materials”.

Caption to Fig. 2: “the pentagon-involved phases”

Caption to Fig. 3: “crystallines” -> “crystallites”

The two concluding paragraphs are disconcerted and incoherent.

It is unclear what structures do the dashed curves in Fig. 2 correspond to; the legend only pertains to solid curves whereas the caption says that “the dashed curves are for the other simple phases”. In the text and in the figures, both “Chengtu” and “Chengdu” are used without an evident reason. I imagine that a single name would be better.

Reviewer #4

(Remarks to the Author)

The authors have done a lot of good work to improve the manuscript, compared to the previous version I saw. There are some valuable interesting results presented in this manuscript. However, there are still a few things that I'm less keen on.

Firstly, the things I like: I very much like the work they have done underpinning Fig.1, Fig.2 and Fig.3a. These results now convince me that the quasicrystals (QCs) observed in Ref.[25] are not in fact the equilibrium low-temperature T phases formed by this system. This is an important result. The results in this manuscript pose a bunch more questions relating to what is the temperature range over which the QCs are stable and how does the phase diagram evolve as T is varied? However, I realise such questions are beyond the scope of the methods used in this work. Given where [25] was published, it seems appropriate to me that the results in this manuscript should appear in the same journal.

Now the things I don't like:

1. I strongly disagree with the sentences "the Hertzian potential energy describes the interaction between two soft-core colloid particles" (in the abstract) and "The Hertzian interaction ... is a prototypical interaction of a soft-core potential" (in the introduction). I think that is overselling this particular pair-potential form and is far too sweeping and strong a statement. I would phrase it something like "the Hertzian potential is a simple model for soft-core colloidal interactions". Be aware that, for example, Gaussian, Yukawa and other forms are used, so giving the impression that the Hertzian form fully describes soft-core colloids is misleading.

2. The connection with the architectural tiling is in my opinion overdone. It is nice to point out these structures are found in architecture and to make the cultural connections. But to have it as a central theme and something that is a thread all through the whole article is in my opinion a distraction from the interesting science. Or to put it another way: I would rather the words dedicated to explaining the architectural connections were spent explaining more the science.

I realise point 2 above is somewhat a matter of "taste" or "beauty", so I'm open to the authors disagreeing with me on this, but on point 1, I feel I must insist.

Version 2:

Reviewer comments:

Reviewer #4

(Remarks to the Author)

Dear Editor,

The authors have addressed all of the points I raised in my previous review and I am now happy to recommend publication.

You asked me to comment on the "ongoing disagreement between the authors and Reviewer #2, who disagrees with the authors on the strength of the conclusions that can be drawn, mainly regarding the applicability of the Hertzian model."

In my opinion the paper does not over-sell the Hertzian model. Reviewer #2 is right that it does not directly model a particular physical system. However, for me that is not a problem. I like the approach taken, to analyse a simple model, because what one learns from studying simple models is often more widely applicable, albeit not applicable to a specific case. In my opinion, the Hertzian potential is a suitable generic model for soft particle.

REPLY to Referee-1

The referee said: "...I feel that it can be published in Nat. Commun. after the authors properly address the following questions."

1. "Which figure/paragraph support the statement "we discovered crystal patterns built hierarchically behind quasicrystals" in the abstract?"

REPLY: The paragraphs, Lines 116-149.

2. "all have a lower system energy than that of the previously proposed QC state". However, I cannot find the value of the energy of the DDQC. Although it cannot be analytically calculated due to its aperiodic nature, the author should provide the measured energy from the simulation to demonstrate that it is indeed higher than the energies of the other five types of quasicrystals."

REPLY: This is a very good question. We have now added Fig. 3A and rewrote the entire last section (Lines 162-216) to address this.

3. By heating a ground state to the low temperature used in ref.25, does the ground state transform to DDQC?

REPLY: Yes. The diamonds in Fig. 3A transitions to circles.

4. "since the very beginning of the discovery (17, 18)" Refs.1,2 are the very beginning of the discovery, why cite 17,18?

REPLY: Apologies. 17 and 18 have been moved to the end of the sentence.

5. "A reference should be added after "... in terms of [α rho]" in page 1."

REPLY: After the revision, that sentence no longer needs reference.

Reviewer #2 (Remarks to the Author):

The manuscript NCOMMS-24-02433-T "Premodern Architectural Tilings behind Hertzian Quasicrystals" by Li, Ye, and Chen, which reports a numerical study of 2D crystalline and quasicrystalline structures formed by particles characterized by the Hertzian pair interaction, cannot be recommended for publication for several reasons as elaborated below, and I suggest it be rejected.

1. “Firstly, the manuscript is hardly novel enough to be suitable for a high-impact journal such as Nat. Commun. The model explored has been studied in the past (e.g., in Ref. [31] and in related papers by Ryzhov, Fomin, Tsiok, and Gaiduk as well as by other authors). Li, Ye, and Chen do seem to extend the existing insight but even if they do so, their results are incremental”

REPLY: There may be some misunderstanding here. Yes, the model has been studied in the past, but using it to explore the formation of pentagon-related crystalline structures in connection with quasicrystals, through an exact treatment that allows for precise determination of the phase diagram, is completely new. The references mentioned, for example, concern different aspects of crystal formation that are unrelated to the pentagon structures studied here.

2. “Secondly, my concern behind the wording “seem to extend” in the previous sentence pertains to the possibility that the analysis in NCOMMS-24-02433-T may not be scientifically sound. It appears to me that the authors constructed the phase diagram in the (α, ρ) plane in Fig. 1A by identifying the minimal-energy structures rather than by considering the complete phase equilibrium, that is by the Maxwell double-tangent construction. I would imagine that the phase diagram in the (α, ρ) plane should feature finite coexistence regions (especially in view of the very small energy differences reported in Fig. 2) but they seem to be absent. If the authors indeed did not do the double-tangent construction, it is possible that some of the claimed stable structures would not make it in the correct phase diagram.”

REPLY: This is a very good point. Our original intention was to consider only the ground states (hence the old Fig. 1A was labeled as a ground-state diagram), with the discovered state contrasted against related papers where ground states were also considered. In this revision, we have transitioned to a true phase diagram, where the first-order transition lines are determined using the double-tangent construction. The revision preserves the overall structure of the original state diagram but enriches it by including coexistence corridors along the phase boundaries. Please see the revised Fig. 1A. We have also updated many parts of the text where "ground state" was originally referenced. We are deeply indebted to the referee for this insightful and helpful suggestion. The main physics remains the same.

3. Thirdly, I do not share the authors’ view that the Hertzian interaction is a good model for the pair potential between soft particles. The pairwise additive Hertzian interaction applies to contact repulsion between elastic bodies such as spheres but only for small indentations, that is for r/σ not much smaller than unity. The authors use it for all

r's, that is well beyond the applicability of this model to real elastic bodies. In addition, it is questionable that soft colloidal particles can be described by the Hertzian potential in the first place as they are too small for the theory of elasticity to be relevant. These concerns make NCOMMS-24-02433-T less relevant for real colloidal systems.

REPLY: The debate over whether the Hertzian model is valid for elastic particles lies beyond the scope of the current manuscript. In a separate study, we examined another well-documented pair potential and found that similar pentagon-related structures, as reported here, can also emerge. Albeit in different regions of the parameter space specific to that model, the pentagon-related crystallines are more common than one would think.

It is worth noting that the term “soft colloidal particles” is used here in a general sense; for example, they may refer to polymeric soft spheres of significantly larger size (see Ref. 21, 22 and 26 in the text), which have been modeled using the Hertzian potential. Notably, in Ref 26 (Grillo et al, Nature 2020) Hertzian model directly reproduced the experimental results. Indeed, recent studies have reported quasicrystal formation in polymer complexes. The question of whether “soft colloidal particles can be described by the Hertzian potential in the first place, given that they are too small for continuum elasticity to apply,” is itself open to debate. In this study, we use the Hertzian model, easy to adopt and conceptually simple, as a tool to explore the formation of pentagon-related structures. Our intention is not to apply the model strictly to the specific class of colloids the referee may have had in mind.

4. “Finally, the presentation of the results is not commensurate with the results themselves; in brief, the results are oversold. The references to architecture, art, history, etc. are superficial and hardly needed.”

REPLY. These references, depending on the audience and community, may hold value. For example, the Cairo tiling has traditionally been associated with a specific type of pentagonal tiling. While the scientific content of the manuscript would remain intact without these references, we believe that they offer useful context. We are open to removing them if necessary but would prefer to seek guidance from the journal's Editor on this matter.

REPLY to Referee-3.

While the referee raised some concerns, they expressed “I truly enjoy reading the manuscript.”

1. “The primary outcome revolves around comparing the ground state energies of the new tilings. Regarding free energies, the Frenkel-Ladd method proves to be valuable for analyzing periodic crystals and has previously been applied to colloidal systems.”

REPLY: The Frenkel-Ladd method deals with the construction of the free energy where the entropy is a component. Here, we mostly work on the energy minima where no entropy is a concern (with the exception of DDQC, studied by MD simulations in the last part).

2. “By using numerical simulations, the authors have not obtained any single phases of new tilings. They may depend on boundary conditions. I recommend that much numerical evidence supporting the main text be included in the Supporting Materials.”

REPLY: MD is actually a poor method to produce single phase structures. The main text is supported by analytic (exact) treatment of the new phases, not MD. The revised sentences, Line 161-182, have effectively addressed this, by explaining that MD will be trapped into co-existing structures. This is common, regardless of the boundary conditions. We could include simulation snapshots of the trapped states in the SM, but feel they do not produce any new information.

3. “In the case of these complex crystal lattices, I presume that the minimization processes are multidimensional. However, the explicit process, including the number of parameters considered, is not delineated in the supporting materials. It may be assumed that symmetry considerations play a role in the process.”

REPLY: The requested information is now available in SM, where the exact variables used are listed for each of the five lattices.

4. “The Cairo tiling is well-known; however, I disagree with the authors' choice of terminology for the Chengtu and Agra lattices, despite their careful justification. To my knowledge, the names Chengtu and Agra lattices are not commonly used to refer to the lattice structures presented in the paper. It should be noted that the Chengtu tiling depicted in Fig. 1C appears to consist of equilateral tiles, including an equilateral triangle and an irregular but equilateral pentagon. However, the pattern described in Ref. 39 (Dye, Fig. C 5b) does not appear to be equilateral. Consequently, I am hesitant to adopt this terminology, even though the wallpaper group symmetry is consistent. Additionally,

for the Agra tiling shown in Fig. 1E, I was unable to locate a reference. While I acknowledge the abundance of beautiful patterns in Agra, I am uncertain about using the Agra lattice to describe this specific and highly complex pattern. Instead, I suggest that the authors consider employing terminology from the theory of mathematical tiling.”

REPLY: We understand the referee’s concern. However, most historical patterns lack the perfect symmetry that we see in these crystallines. The use of Chengtu, for example, is based on the cells are built into the pattern. We revised the reference to Chengtu in line 106: “... a similar pattern can be seen on the armor of door gods and in window grilles in Chengdu, China, although these patterns lack perfect p31m symmetry.” The reference to Agra pattern is according to a pattern, documented in Ref. [41,42]. Please also see line 120 for the revision: “many other decorative patterns also exist in the Agra tradition, this particular one matches the superlattice found here.”

We considered opto the terminology in math tiling but then feel it may build barrier for general readers. We have maintained the “Pen-Sq”, “PenTri-Tri”, and “PenHex-Sq” names, together with these nicknames, as they are more descriptive.

5. “The authors made assumptions regarding models, including the Cairo, Chengtu, PenHex-Str, Agra, and PenTri-Rec tilings. However, these assumptions lack systematic consideration. I would suggest that the authors consult the theory of tilings for a more rigorous foundation. Alternatively, the hierarchical or stepwise construction employed in this study recalls the approximant phases of quasicrystals.”

REPLY: These states were discovered in two ways. (1) by lowering the simulated temperature from about 1,000 MD simulations and examining the resultant diffraction patterns of the phase separated states. (2) by systematically considering the tiling patterns in the theories of Ref. (37) and (38). We believe that we have done an exhaustive search. See line 102.

6. “For the names of the wallpaper (plane) group, the authors use the shorter versions. It would be better to use the formal (IUCr) version; say “p4g” is “p4gm.” You may find the correspondence in a table in Wikipedia “wallpaper group”.”

REPLY: Very good suggestions. Revised!

7. “The Chengtu tiling has been deeply studied in a PRL (2011) paper in the context of colloidal particle simulations. DOI: 10.1103/PhysRevLett.106.095504” ,
REPLY. Thanks! Our ignorance! That reference is now cited as Ref. 40.

8. Figure caption Fig.3 C Cairo and PenHex-Str? -> Cairo-Chengtu.

REPLY: Thanks for the carefulness. We have now changed the reference.

9. PenHex-Str (Fig) vs PenHex-Rec in the main text (L124, L133).

REPLY. Thanks! They have been revised.

10. L.163 3D -> 3C

REPLY: Thanks! Fixed.

REPLY to referee 4.

The referee praised the manuscript, “...This is an important finding, if true...”. Then he raised a few concerns.

(1) “However, I'm concerned that the way the energies for each structure are calculated is artificially penalising the candidate quasicrystalline structures that are actually observed at finite temperature. The energy of a quasicrystalline structure depends significantly on the box size in which one calculates. Remember, strictly speaking, one can never find a quasicrystal in a finite size box. There are methods to get around this for practical calculations. One approach is to work with a higher-dimensional periodic systems and then cut and project to obtain the real lower dimensional system. Typically, one looks at a 4-dimensional periodic structure to calculate for the real 2-dimensional system. Another approach is to consider what finite sized box leads to a structure which forms a good finite size approximant to the true infinite quasicrystal. I saw no discussion about varying the size of the box used for the simulations of the quasicrystal nor any discussion of how the box size was chosen. I'd also like to see how the quasicrystal free energy varies for different choices of box when plotted on top of the curves displayed in Fig.2. Is the quasicrystal energy really nowhere near the energy of the periodic structures displayed?”

REPLY. The referee's main concern is what happens in energy comparison. We have taken this seriously and have rewritten the entire latter part of the paper, including the introduction of a new figure, Fig. 3A, where the QC energy is overlaid on top of Fig. 2C.

Essentially, sufficiently low temperatures favor a QC, but even lower temperatures lead to crystalline structures. Yes, QC is nowhere near the energy basins of Fig. 3A.

The other concern is the role of boundary conditions. By varying the size of the simulation box, the boundary effects are minimized, if not completely removed. Two points are noted: (a) When plotting the energies of the circles, squares, and diamonds in Fig. 3A, we estimated that the error bars due to boundary conditions are less than the symbol size. (b) Our main focus is how QC content varies as a function of temperature, not whether the entire simulated box can reach 100% QC (which perhaps require 4D analysis). Displayed by the gray bar in Fig. 3A, the content cannot be 100% in a 2D system. Please also see the last paragraph of section-2 of Method, line 421.

- (2) “While I agree the connection to patterns in architecture is interesting, this has been discussed before. I feel that the connections between the model pair potential and the particular structures formed and why they arise in historical building decorations needs to be drawn out better.”

REPLY: Yes, the Cairo similarity was indeed discussed before. The Chengtu and Agra comparison is new. Historical building decorations had no concerns on the pair potential energy used here, of course, as they belong two different camps of intellectual development. We brought the comparison in, to this paper, simply because of cultural interest and we are afraid that we cannot make a deeper connection between the two.

- (3) “The correlation functions $g(r)$ in Fig.3 are interesting but the text does not really explain why these particular examples have been selected, nor what we are supposed to learn from these. These need to be discussed more. Ideally, I'd want to see how they vary as parameters in the model are varied.”

REPLY: We introduce FIG 3, for the purpose of answering the question of “is 12-fold diffraction symmetry unique to DDQC”. We have rewritten the paragraph near the end of page 3, to sharpen the message more clearly. “how they vary” depends on how MD is done and what structures are trapped and there is no definitive answer. Because the main focus of the paper is exact treatment, expanding MD might obscure the focus. Nevertheless, we revised the next sentence to “Over most of the colored phase regions of Fig. 1, MD runs are typically trapped energetically to contain pentagon-related crystalline domains in different orientations ...”, implying this is common in MD.

- (4) “A final, minor point: I don't really understand the use of the word "behind" in the title.”

REPLY: a good question. We have now changed it to “Premodern Architectural Tilings and Hertzian Quasicrystals”.

Reply to reviewers' comments.

Reviewer #1:

The referee commented. "The authors have addressed my questions and comments satisfactorily. However, the caption of the newly added Fig. 3A should be improved, e.g., explaining the meaning of the solid and dashed curves."

REPLY: Thanks. A sentence has been added to the caption to explain these curves.

Reviewer #4:

The reviewer commented: The authors have done a lot of good work to improve the manuscript, compared to the previous version I saw. There are some valuable interesting results presented in this manuscript. However, there are still a few things that I'm less keen on. Firstly, the things I like: I very much like the work they have done underpinning Fig.1, Fig.2 and Fig.3a. These results now convince me that the quasicrystals (QCs) observed in Ref.[25] are not in fact the equilibrium low-temperature T phases formed by this system. This is an important result. The results in this manuscript pose a bunch more questions relating to what is the temperature range over which the QCs are stable and how does the phase diagram evolve as T is varied? However, I realise such questions are beyond the scope of the methods used in this work. Given where [25] was published, it seems appropriate to me that the results in this manuscript should appear in the same journal.

1. The referee said: "I strongly disagree with the sentences "the Hertzian potential energy describes the interaction between two soft-core colloid particles" (in the abstract) and "The Hertzian interaction ... is a prototypical interaction of a soft-core potential" (in the introduction). I think that is overselling this particular pair-potential form and is far too sweeping and strong a

statement. I would phrase it something like "the Hertzian potential is a simple model for soft-core colloidal interactions". Be aware that, for example, Gaussian, Yukawa and other forms are used, so giving the impression that the Hertzian form fully describes soft-core colloids is misleading.

REPLY: The disagreed sentences, in the abstract and introduction have been rewritten. Please see the first sentence in the abstract, "As a simple physics model supporting the current understanding of crystal and quasicrystal formation, the Hertzian potential...", and in the introduction "...Hertzian potential... one of the simplest models of a soft-core potential...". The second paragraph has also been softened, to avoid repeating the word "challenging".

2. The referee said: " The connection with the architectural tiling is in my opinion overdone. It is nice to point out these structures are found in architecture and to make the cultural connections. But to have it as a central theme and something that is a thread all through the whole article is in my opinion a distraction from the interesting science. Or to put it another way: I would rather the words dedicated to explaining the architectural connections were spent explaining more the science. I realise point 2 above is somewhat a matter of "taste" or "beauty", so I'm open to the authors disagreeing with me on this, but on point 1, I feel I must insist.

REPLY: In this revision, we have generally toned down the historical and artistic comparisons. The relevant references in the title, abstract, introduction, and summary have all been removed. Only one condensed paragraph remains, near the end of page 1, simply to note where the nicknames of the structures originate.

Reviewer #2 (Remarks to the Author):

Reviewer #2:

The referee opened by saying “I have read the revised version of NCOMMS-24-02433-A “Premodern Architectural Tilings and Hertzian Quasicrystals” by Li, Wang, Geng, Liu, Ye, and Chen as well as the rebuttal letter. The authors made some changes to the manuscript but my reservations remain. I do not find the manuscript suitable for publication in Nat. Commun. and I suggest that it be rejected. The manuscript NCOMMS-24-02433-T “Premodern Architectural Tilings behind Hertzian Quasicrystals” by Li, Ye, and Chen, which reports a numerical study of 2D crystalline and quasicrystalline structures formed by particles characterized by the Hertzian pair interaction, cannot be recommended for publication for several reasons as elaborated below, and I suggest it be rejected.”

Regarding original comment 1.

1. Commented in round-1: “Firstly, the manuscript is hardly novel enough to be suitable for a high-impact journal such as Nat. Commun. The model explored has been studied in the past (e.g., in Ref. [31] and in related papers by Ryzhov, Fomin, Tsiok, and Gaiduk as well as by other authors). Li, Ye, and Chen do seem to extend the existing insight but even if they do so, their results are incremental”

REPLY in round-1: There may be some misunderstanding here. Yes, the model has been studied in the past, but using it to explore the formation of pentagon-related crystalline structures in connection with quasicrystals, through an exact treatment that allows for precise determination of the phase diagram, is completely new. The references mentioned, for example, concern different aspects of crystal formation that are unrelated to the pentagon structures studied here.

New comments in round-2: “The question is whether the paper contains enough completely new results to warrant publication in a high-impact journal. In my opinion, the crystal structures that contain pentagonal motifs and the dodecagonal quasicrystal reported are a modest step ahead. Indeed, it is possible that crystal lattice containing pentagons were not reported in a 2D system of Hertzian particles so far but not all of them are completely new. For example, the PenHex-Str structure was discussed 27 years ago in 2D particles with hard core + linear ramp potential [E. A. Jagla, Phys. Rev. E. 58, 1478 (1998); structure S4 shown in Fig. 2 of Jagla’s paper]; in this paper, the PenHex-Str lattice is located right next to the stretched honeycomb (Shon) and the stripe (Str) lattice, and not far from the Square (Sq) lattice – just like in NCOMMS-24-02433-A. While the Hertzian potential is different from the potential used by Jagla in the above reference, they share the soft repulsive nature and so is it not surprising that they lead to a similar phase diagram. The “exact treatment that allows for precise determination of the phase diagram” mentioned by the authors in the rebuttal is not completely new either. In 1998, Jagla used the same approach but he noted that “other (more stable) structures may have been missed”. The structures reported in NCOMMS-24-02433-T may be some of those missed by Jagla. At the same time, it is quite possible that Li et al. may have missed one or more complex structures themselves. – Yet another old reference with crystals that include pentagonal motifs is A. Skibinsky et al., Phys. Rev. E 60, 2664 (1999); the potential used here is isotropic hard core + square well. I imagine that there exist other related references both in the field of colloids and in other fields, possibly older than Jagla 1998 and Skibinsky et al. 1999. As per the quasicrystals reported in NCOMMS-24-02433-T at finite temperatures, I note that Jagla’s paper also contains a decagonal quasicrystal containing many regular pentagonal and stretched honeycomb local environments. The same is true for the Skibinsky et al. paper. Hence my view that the results reported in NCOMMS-24-02433-T do not constitute a novel enough and important enough body of work to warrant publication in Nat. Commun.”

REPLY: Both references have now been added to the manuscript and discussed. Both work of Jagla and Skibinsky are included in Lines 113–119, which reported the mentioned structure through computer simulations. The added paragraph, before “in summary”, now discusses these references (and others). Please refer to page 5, right column.

It should be noted that the main theme of this paper is not a competition to report new structures (though some of ours are indeed new). A key finding of our study is the analytical evidence we provide showing how complicated hierarchical structures can emerge in the region where quasicrystals may appear. Then, the computer simulations on how quasicrystal emerge above these. The existence of a previously reported unit cell does not weaken our work; rather,

it reinforces the central theme of this paper: that analytical evidence supports a potential model for the quasicrystal hierarchy hypothesis.

Finally, we have added other concluding sentences in the Summary to reflect the main theme of our study [how universal is hierarchy in soft potential, and what are the basic unit cells?].

Regarding original comment 2.

2. The referee commented in round-1: “Secondly, my concern behind the wording “seem to extend” in the previous sentence pertains to the possibility that the analysis in NCOMMS-24-02433-T may not be scientifically sound. It appears to me that the authors constructed the phase diagram in the (α, ρ) plane in Fig. 1A by identifying the minimal-energy structures rather than by considering the complete phase equilibrium, that is by the Maxwell double-tangent construction. I would imagine that the phase diagram in the (α, ρ) plane should feature finite coexistence regions (especially in view of the very small energy differences reported in Fig. 2) but they seem to be absent. If the authors indeed did not do the double-tangent construction, it is possible that some of the claimed stable structures would not make it in the correct phase diagram.”

REPLY in round-1: “This is a very good point. Our original intention was to consider only the ground states (hence the old Fig. 1A was labeled as a ground-state diagram), with the discovered state contrasted against related papers where ground states were also considered. In this revision, we have transitioned to a true phase diagram, where the first-order transition lines are determined using the double-tangent construction. The revision preserves the overall structure of the original state diagram but enriches it by including coexistence corridors along the phase boundaries. Please see the revised Fig. 1A. We have also updated many parts of the text where “ground state” was originally referenced. We are deeply indebted to the referee for this insightful and helpful suggestion. The main physics remains the same.”

The referee commented in round-2: “What thermodynamic potential was used for the double-tangent construction? Enthalpy or energy? Or was it the energy less the energy of the triangular lattice shown in Fig. 2? In my experience, the small energy differences shown in Fig. 2 (e.g., between Cairo in Agra at $\alpha = 2.5$) usually lead to broad regions of coexistence. From the data included in the manuscript, it is unclear how the authors constructed the phase diagram.

REPLY: Given the energies per unit area, there is only one valid way to perform the double-tangent construction. In this revision, we have added a section to the SM to clarify any potential confusion. Since our calculations are based on analytical solutions, there is no ambiguity.

The referee continued in round-2: “Fig. 2 still shows the range of densities where phases that include pentagons have the lowest energy (“The shaded ranges highlight the approximate density where pentagon-involved phases are preferred”), which is misleading.”

REPLY: To avoid confusion, we have rephrased that sentence to “The shaded ranges highlight the approximate density regions where pentagon-involved phases are identified through a double-tangent construction”.

Regarding original comment 3.

3. The referee commented in round-1 “Thirdly, I do not share the authors’ view that the Hertzian interaction is a good model for the pair potential between soft particles. The pairwise additive Hertzian interaction applies to contact repulsion between elastic bodies such as spheres but only for small indentations, that is for r/σ not much smaller than unity. The authors use it for all r ’s, that is well beyond the applicability of this model to real elastic bodies. In addition, it is questionable that soft colloidal particles can be described by the Hertzian potential in the first place as they are too small for the theory of elasticity to be relevant. These concerns make NCOMMS-24-02433-T less relevant for real colloidal systems.”

REPLY in round-1: The debate over whether the Hertzian model is valid for elastic particles lies beyond the scope of the current manuscript. In a separate study, we examined another well-documented pair potential and found that similar pentagon-related structures, as reported here, can also emerge. Albeit in different regions of the parameter space specific to that model, the pentagon-related crystallines are more common than one would think.

It is worth noting that the term “soft colloidal particles” is used here in a general sense; for example, they may refer to polymeric soft spheres of significantly larger size (see Ref. 21, 22 and 26 in the text), which have been modeled using the Hertzian potential. Notably, in Ref 26 (Grillo et al, Nature 2020) Hertzian model directly reproduced the experimental results. Indeed, recent studies have reported quasicrystal formation in polymer complexes. The question of whether “soft colloidal particles can be described by the Hertzian potential in the first place, given that they are too small for continuum elasticity to apply,” is itself open to debate. In this study, we use the Hertzian model, easy to adopt and conceptually simple, as a tool to explore the formation of pentagon-related structures. Our intention is not to apply the model strictly to the specific class of colloids the referee may have had in mind.

The referee commented in round-2: “In the first sentence of the abstract, the authors say “In its unique place as a fundamental physics model and corner stone supporting the current understanding of formation of crystals and quasicrystals, the Hertzian potential energy describes the interaction between two soft-core colloid particles.” In their response, the authors

say that the relevance of the Hertzian model is “open of debate”, which is evidently inconsistent with the first sentence of the abstract. In my view, the opening sentence of the abstract is exaggerated, because the Hertzian potential is not a cornerstone of the current understanding of formation of crystals and quasicrystals. Equally strong is the authors’ claim that the Hertzian interaction is “the prototypical interaction of a soft-core potential” (l. 31-33). I beg to differ. Why should the Hertzian interaction be the prototype? It seems to me that these statements are included so as to elevate the importance of the authors’ work; in my opinion, these statements are unjustified.

REPLY: We take the referee’s critique seriously and have toned down the impression that the Hertzian model is a cornerstone. The mentioned sentences have been replaced with more moderate statements. Please see the highlighted revisions on page 1: “As a simple ...” and “provides one of the simplest ...”

The referee continued in round-2: “The Hertzian interaction is derived by assuming that the indentation of the particles in contact is small and that the deformation of the particles is described by the harmonic theory of elasticity. None of this holds at large indentations, and there exists no theoretical justification for the use of the Hertzian theory at indentations beyond a few percent. I do not see how this can be open to debate. In addition, the size of colloidal particles is typically no larger than a micrometer or else they sediment. It is hardly evident that particles as small as this can be regarded as elastic continua like macroscopic entities such as cm-size rubber balls.”

REPLY: The referee’s statement is essentially debating the validity of the Hertzian model. This is why, in our round-1 response, we already noted politely that “the relevance is open to debate.” The reference to “harmonic theory” in the referee’s comment is not accurate. With α as a varying parameter, $\alpha = 2$ (harmonic) is only one special case, whereas most of the interesting states appear at larger α ’s. Whether these larger α values hold at large indentations, and whether the referee’s “a few percent” statement is justified, remain unsubstantiated in his reasoning. The referee is again raising this point without firm supporting evidence.

We note that the use of continuum models at the micrometer scale is well established in other fields, such as polymer theory, where polymers of comparable conformational size (approximately micrometers) are routinely described by continuum models. An extensive research community has developed around such approaches. For this reason, we suggest setting aside further debate on the applicability of continuum theory at this scale, as the issue is about the entire research community, lying beyond the scope of the present paper.

Regarding original comment 3.

4. The referee commented in round-1: “Finally, the presentation of the results is not commensurate with the results themselves; in brief, the results are oversold. The references to architecture, art, history, etc. are superficial and hardly needed.”

REPLY in round-1: “These references, depending on the audience and community, may hold value. For example, the Cairo tiling has traditionally been associated with a specific type of pentagonal tiling. While the scientific content of the manuscript would remain intact without these references, we believe that they offer useful context. We are open to removing them if necessary but would prefer to seek guidance from the journal’s Editor on this matter.”

The referee commented in round-2: “A considerable part of the manuscript, especially in l. 95-123, refers to patterns used in architecture and for decoration. These references are episodic and serve no scientific purpose (e.g. “Dating back as early as 1800 A.D., the Chengtu pattern was scholarly documented by physicist D. S. Dye in his 1937 book (39), which recorded over a thousand types of Chinese lattice designs”); no conclusion is drawn based on the comparisons made. In their rebuttal, the authors say this themselves (“... the scientific content of the manuscript would remain intact without these references ...”). I do not share the authors’ view that “[these references] offer useful context”. They may appeal to a general audience and would possibly fit into a popular science article. In a research paper, on the other hand, they are unneeded or may even be a distraction. In a similar vein, I find the title “Premodern architectural tilings and Hertzian quasicrystals” misleading. In the manuscript, the authors do not establish a connection between the architectural tilings and 2D structures formed by Hertzian particles (in my opinion, this cannot be done because such a connection does not exist). What is then the purpose of the title, which makes one expect the connection? The last sentence of the abstract reads “The connection between art history and the new crystal patterns can mutually stimulate the creativity of art designs and new material designs.” What exactly did the authors mean by this? Firstly, this connection does not exist, and secondly, if they meant the similarity of decorative and crystal patterns, this is no news; after all, 2D space groups are referred to as the *wallpaper* groups. There is no need to emphasize this in the abstract of a paper in Nat. Commun. in 2025.”

REPLY:

In this revision, we have generally toned down the historical and artistic comparisons. All relevant references in the title, abstract, introduction, and summary have been removed. Only a single condensed paragraph remains, near the end of page 1, solely to indicate the origins of the structures’ nicknames.

OTHER comments:

The referee commented in round-2 :“The language is often poor – here are some examples...”

I. 193-185: “The pentagon-related crystals and DDQC are all composed of different mixing of pentagons and other geometric identities, in various tiling patterns.” -> “The pentagon-related crystals and DDQC are all composed of different mixture of pentagons and other geometric entities, in various tiling patterns.”

REPLY: Thanks! We have revised that sentence.

I. 204-205: “underlining relationship” -> “underlying relationship”

REPLY: Revised. Thanks!

I. 209: “the radius distributions” -> “the radial distribution function”

REPLY: Revised. Thanks!

I. 224-225: “can stimulate the creativity of new material design” -> “can stimulate the design of materials”.

REPLY: That sentence has been removed.

Caption to Fig. 2: “the pentagon-involved phases”

REPLY: no errors found here.

Caption to Fig. 3: “crystallines” -> “crystallites”

REPLY: Revised. Thanks!

The two concluding paragraphs are disconcerted and incoherent.

REPLY: The two paragraphs are now condensed into one, removing the cultural reference and with minor revisions.

The referee commented: "It is unclear what structures do the dashed curves in Fig. 2 correspond to; the legend only pertains to solid curves whereas the caption says that "the dashed curves are for the other simple phases". In the text and in the figures, both "Chengtu" and "Chengdu" are used without an evident reason. I imagine that a single name would be better.

REPLY: The dashed lines are the simple structures listed in FIG 1. We have added legend directly in the figure, to repeat the information.